# Endogenous Retrovirus RNA Expression Differences between Race, Stage and HPV Status Offer Improved Prognostication among Women with Cervical Cancer

**DOI:** 10.3390/ijms24021492

**Published:** 2023-01-12

**Authors:** Jill Alldredge, Vinay Kumar, James Nguyen, Brooke E. Sanders, Karina Gomez, Kay Jayachandran, Jin Zhang, Julie Schwarz, Farah Rahmatpanah

**Affiliations:** 1Department of Obstetrics and Gynecology, University of Colorado, Aurora, CO 80045, USA; 2Department of Pathology and Laboratory Medicine, University of California, Irvine, CA 92697, USA; 3Department of Radiation Oncology, Washington University School of Medicine, St. Louis, MO 63108, USA; 4Siteman Cancer Center, Washington University School of Medicine, St. Louis, MO 63108, USA; 5Institute for Informatics, Washington University School of Medicine, St. Louis, MO 63108, USA; 6Department of Cell Biology and Physiology, Washington University School of Medicine, St. Louis, MO 63108, USA

**Keywords:** human endogenous retroviruses, cervical cancer, HPV, ethnic disparity, prognostic model

## Abstract

Endogenous human retroviruses (ERVs) are remnants of exogenous retroviruses that have integrated into the human genome. Using publicly available RNA-seq data from 63 cervical cancer patients, we investigated the expression of ERVs in cervical cancers. Four aspects of cervical cancer were investigated: patient ancestral background, tumor HPV type, tumor stage and patient survival. Between the racial subgroups, 74 ERVs were significantly differentially expressed, with Black Americans having 30 upregulated and 44 downregulated (including MER21C, HERV9-int, and HERVH-int) ERVs when compared to White Americans. We found that 3313 ERVs were differentially expressed between HPV subgroups, including MER41A, HERVH-int and HERVK9. There were 28 downregulated (including MLT1D and HERVH-int) and 61 upregulated (including MER41A) ERVs in locally advanced-stage compared to early-stage samples. Tissue microarrays of cervical cancer patients were used to investigate the protein expression of ERVs with protein coding potential (i.e., HERVK and ERV3). Significant differences in protein expression of ERV3 (*p* = 0.000905) were observed between early-stage and locally advanced-stage tumors. No significant differential expression at the protein level was found for HERVK7 (*p* = 0.243). We also investigated a prognostic model, supplementing a baseline prediction model using FIGO stage, age and HPV positivity with ERVs data. The expression levels of all ERVs in the HERVd were input into a Lasso-Cox proportional hazards model, developing a predictive 67-ERV panel. When ERVs expression levels were supplemented with the clinical data, a significant increase in prognostic power (*p* = 9.433 × 10^−15^) relative to that obtained with the clinical parameters alone (*p* = 0.06027) was observed. In summary, ERV RNA expression in cervical cancer tumors is significantly different among racial cohorts, HPV subgroups and disease stages. The combination of the expression of certain ERVs in cervical cancers with clinical factors significantly improved prognostication compared to clinical factors alone; therefore, ERVs may serve as future prognostic biomarkers and therapeutic targets. Novelty and Impact: When endogenous retroviral (ERV) expression signatures were combined with currently employed clinical prognosticators of relapse of cervical cancer, the combination outperformed prediction models based on clinical prognosticators alone. ERV expression signatures in tumor biopsies may therefore be useful to help identify patients at greater risk of recurrence. The novel ERV expression signatures or adjacent genes possibly impacted by ERV expression described here may also be targets for the development of future therapeutic interventions.

## 1. Introduction

Cervical cancer affects millions of women worldwide and has the highest incidence amongst gynecologic cancers. Human papilloma virus (HPV) is a sexually transmitted DNA virus responsible for almost all cervical cancers, with HPV 16 and 18 subtypes contributing to 70% of the cases [1]. The CDC estimates that in the United States alone, there are 21,100 new cases of HPV-associated cancer in women each year, the majority of these cases being cancers of the cervix, vulva and vagina [2]. The E6 and E7 genes in the HPV DNA both play key roles in carcinogenesis. The E6 protein inhibits the p53 tumor suppressor gene, while the E7 protein inactivates the Rb tumor suppressor protein, thereby interrupting cell cycle regulation [3]. The influence of HPV in cervical cancer progression suggests the potential role of epigenetic modulation in mediating the integration of the viral genome into host DNA and in facilitating unique transcriptome expression differences, a topic which remains yet unstudied.

Endogenous retroviruses (ERVs) are remnants of exogenous retroviruses that integrated into human germline cells, propagating vertically, allowing them to assimilate with the mammalian genome over millions of years. They comprise approximately 8% of the human genome, as compared to the 1–2% responsible for protein-coding genes [4,5]. While most ERVs remain inactive, emerging research suggests that a minority have become activated and that these ERVs may play a role in epigenetic modification through DNA methylation or histone modification, resulting in gene regulation or transcriptional alteration [6,7]. This study explored the novel expression profiles of ERVs within cervical cancers.

The disease course of cervical cancer is significantly different between races, and multiple studies suggest that African American women experience more aggressive forms of cervical cancer and face poorer outcomes compared to their White American counterparts [8], an effect which persists after controlling for socioeconomic factors [8,9]. This highlights the need to understand the biological mechanisms by which these disparities occur. Kumar et al. showed significantly differentially expressed ERVs in Black American (BA) and White American (WA) prostate cancer patients; the translation of this work into the realm of cervical cancer and the role of ERVs in racial disparity remain unstudied [10,11]. Understanding how ERV expression is modulated by ancestral ethnicity is crucial for determining the effectiveness of possible therapeutic targets.

Most ERV sequences have acquired numerous mutations over time and therefore do not have protein-coding potential nor the capability to generate infectious viral particles [12]. However, several ERVs retain a complete open reading frame, allowing for quantification of their protein expression in tissue samples. Furthermore, ERVs have demonstrated to be prognostic biomarkers in other types of cancer [11,13,14,15,16]. Therefore, this study aimed to investigate the association between ERV expression and the risk of cervical cancer recurrence.

Studying ERVs has been limited, given the repetitive nature of these elements in the human genome. We developed a novel mapping pipeline for ERV analysis in addition to a unique computational tool allowing the prediction of ERV expression linked to patient outcomes [11]. Currently, there are several prognostic factors that have been investigated for their impact on survival in cervical cancer patients, including histologic subtype (i.e., adenocarcinoma, squamous cell carcinoma), maximum depth of cervical stromal invasion, longitudinal tumor diameter, lympho-vascular space invasion, parametrial invasion and pelvic lymph node metastases. The use of the standardized uptake value (SUV_max_) in ^18^F-fluorodeoxyglucose positron emission tomography is also a prognostic marker associated with tumor infiltration by immune cells and activation of inflammatory pathways [17]. Combining independent predictors of survival including nodal metastases, histology and tumor diameter into a multi-factorial model for risk stratification is more predictive than utilizing the International Federation of Gynecology and Obstetrics (FIGO) stage to predict survival; however, there are no data on the role of ERVs in modulating the ability to predict recurrence-free or overall survival. Here, we show that when our 67-gene ERV expression prognosticator is applied to cervical cancer patients (n = 62), the risk assessment for relapse improves compared to when using clinical parameters alone.

## 2. Results

### 2.1. ERV Expression Profiles in Cervical Cancer Patients with Different Ancestries

To explore whether ERV expression varies among cervical cancer patients with different geographical ancestries, we analyzed the RNA-seq data from GEO GSE151666 [18]. Of the 68 samples used, the ancestral breakdowns were as follows: 15 patients of primarily African ancestry, 48 of primarily European or Middle Eastern ancestry, and 5 of Asian or Hispanic ancestry. As described previously, we classified those of Middle Eastern ancestry and European ancestry under an aggregate classification of the White cohort because population studies using 1000 genomes revealed a smaller genetic distance between patients with Middle Eastern and European ancestries when compared to those of African descent [11,19]. Patients that had a majority calculated Asian or Hispanic ancestry were excluded from the downstream analysis, resulting in a total of 63 patients analyzed (Table 1). For the purposes of only the FIGO stage comparison, one patient who was not within the stage qualifications was excluded. After alignment, differential expression analysis was performed on count data for ERVs using DeSeq*2* [20]. Of the 519,060 ERVs in the HERVd [21], we identified 74 ERVs with a p_adj_ < 0.1 (Appendix A). A total of 74 ERVs passed this cutoff, with BA patients having 30 upregulated and 44 downregulated ERVs when compared to WA patients (Appendix A, Figure 1). Among the significantly differentially expressed ERVs in BA patients compared to WA were MER21C, HERV9-int, and HERVH-int.

A total of three genes were found to be both concordantly regulated and nearby to the differentially expressed ERVs, including CYP2J2, SYT12 and RARRES1 (Appendix A). CYP2J2 was downregulated in BA, while both SYT12 and RARRES1 were identified as upregulated in BA as compared to WA, similar to their nearby ERVs.

### 2.2. ERV Expression Profiles in Cervical Cancer Patients with Tumors at Different Stages

We used DeSeq2 to analyze the expression of ERVs from the HERVd based on the cervical cancer FIGO stage of the samples. The samples were assigned to either early- or locally advanced-stage classifications based on 2018 FIGO classifications [22]. From the total of 62 patients used for downstream analysis, 16 were classified as early-stage, and 46 were classified as locally advanced-stage (Table 1). Of the 89 ERVs (*p_adj_* < 0.1) found to be differentially expressed between the two stages, 28 were downregulated, and 61 were upregulated in locally advanced-stage compared to early-stage samples (Appendix A, Figure 2). A small subgroup of ERVs have protein-coding potential due to their open reading frames and a few, including HERVK and ERV3, were found to be expressed at significantly different levels, *p* < 0.001 in the RNA-seq data.

When samples of different stages (i.e., locally advanced vs. early-stage) were compared, a total of 15 genes were identified following the same direction of regulation as the ERVs, including RBP7 (↓), SLC44A5 (↑), NMT (↑), CADM1 (↑), TUBB3 (↑), PNMAL1 (↑), MERTK (↑), FN1 (↑), C2orf54 (↓), CSTB (↓), FAM3D (↓), MME (↑), SPINK5 (↓), TUSC3 (↑) and MERTK (↑) (Appendix A). 

### 2.3. ERV Expression Profiles in Cervical Cancer Patients with Different HPV Statuses

The samples were separated into two groups based on whether the corresponding patients had been diagnosed with HPV. A total of 53 patients were classified as HPV-positive, and the remaining 10 were assigned to the HPV-negative category. The ERV expression data for these samples were provided to DeSeq2 for differential expression analysis. Using the HERVd, we determined that 3313 ERVs were differentially expressed between samples of different HPV status (Appendix A, Figure 3). Of the differentially expressed ERVs, 1477 were downregulated, and 1836 were upregulated in HPV-negative patients compared to HPV-positive patients.

Among the genes found to be nearby the significantly differentially expressed ERVs in HPV-negative vs. HPV-positive, were CLCNKA (↑), CADPS (↑), ST6GALNAC5 (↑), NCAM1 (↑), CADM1 (↑), STXBP6 (↑), SHCBP1L (↓), and DHRS2 (↓), which were concordant in their differential expression, suggesting the possibility of a regulatory effect (Appendix A).

### 2.4. Protein Expression Analysis of ERV3 in Cervical Cancer Samples of Different Tumor Stages Using TMAs

To determine if the differential expression of ERVs between samples of different tumor stage was also manifested at the protein level, we further investigated ERV3. This ERV was selected not only because it was one of the most significant (*p* < 0.001) differentially expressed genes in locally advanced vs. early-stage tumors in the RNA-seq analysis, but also because it is one of the few ERVs that retains an open reading frame, allowing for the investigation of its protein expression [12]. An ERV3 antibody was applied to a large tissue microarray of tumor tissues from an independent set of early- (n = 28) and locally advanced-stage (n = 19) tumors, and protein production was scored as percent positive pixels using QuPath [23], as described. Consistent with the RNA-seq data, we found significant differences in protein expression for ERV3 (*p* = 0.000905) in locally advanced- vs. early-stage samples (Appendix A, Figure 4). HERV-K protein expression was also analyzed, but no significant differential expression at the protein level was found (*p* = 0.243).

### 2.5. Recurrence Risk Assessment of Selected ERVs in Cervical Cancer Patients

The ability to successfully classify patients as either high-risk or low-risk for disease progression is of incredible value for patient management. In our previous study [11], we demonstrated that a combination of ERV expression signatures alongside clinical data could provide valuable insights to predict a specific patient’s recurrence probability. Here, we applied a similar methodology, examining the sum effects of ERV expression with cervical cancer specific measurements such as FIGO stage and general patient characteristics including age and HPV positivity.

After an initial preselection step using a univariate Cox model, we used the expression levels of all ERVs in the HERVd as an input for a Lasso-Cox proportional hazards model [24]. The final risk model was built by combining the 67-ERV panel expression data with the clinical information for all patient samples. For a baseline comparison, we built a prediction model using only the clinical variables, including FIGO stage, age and HPV positivity, by fitting a Cox proportional hazards model. The linear predictors from both models separated patients into high- and low-risk groups.

Kaplan–Meier survival analysis completed using only the clinical parameters resulted in a significance level of *p* = 0.06027 (Figure 5A). The full model, which combined the 67-ERV panel expression data and the clinical data, discriminated the two risk groups of patients for time to recurrence at a much higher significance level of *p* = 9.433 × 10^−15^ (Figure 5B). The 67-ERV panel was also internally validated using a 10-fold cross-validation in the glmnet package when the Lasso-Cox proportional hazards model was created.

## 3. Discussion

The role of ERVs in tumorigenesis has been reported in endometrioid cancer, testicular cancer, breast cancer, urothelial cell carcinoma of the bladder, ovarian cancer, clear cell renal cancer and prostate cancer [25,26,27,28,29,30,31,32]. Our investigation into endogenous retrovirus expression profiles in cervical cancer patients allowed for the exploration of the hypothesis of correlations between tumor HPV status, tumor stage and patient ethnicity as well as of nearby genes and the possible influence of ERV expression on their up- or downregulation in the process of carcinogenesis. Among the 74 ERVs differentially expressed between Black Americans and White Americans, the downregulation of both MER21C and HERVH-int were notable in that HERVH has been shown to act alongside the Sp1 protein family to direct transcription of adjacent cellular genes [33]. MER21C is an ERV promotor that in a study by Song et al. can bind to inhibitor of growth protein 3 (ING3), resulting in EZH2-mediated epigenetic trimethylation modification, silencing its expression and preventing innate immune activation [34]. The nearby genes impacted by the other significantly differentially expressed ERVs include CYP2J2, SYT12 and RARRES1. CYP2J2 belongs to the cytochrome P450 family and is a gene of interest in numerous cancer types as a target for therapeutic inhibition [35]. RARRES, another gene nearby significantly differentially expressed ERVs, has implications in carcinogenesis, tumor proliferation, and acts as a tumor suppressor gene by which hypermethylation facilitates metastases in some cancer types [36,37]. SYT12 acts as an oncogene in non-gynecologic cancer types [38]. Racial disparity in cervical cancers is multifactorial in nature, with disparate vaccination, insufficient screening and treatment differences informing worse survival outcomes [39]. The molecular mechanism behind disparate outcomes remains an area of active exploration; however, the differential expression of these ERVs and their role in the epigenetic modulation of oncogenes or tumor suppressor genes could impart subtle differences to carcinogenesis or treatment response between Black and White Americans impacted with cervical cancer [10].

Locally advanced-stage cervical cancers were noted to have 28 downregulated and 61 upregulated ERVs when compared to early-stage cancers, including a small subgroup with protein-coding capacity such as ERV3, which was further validated as under-expressed at the protein level in locally advanced tumors. A significant ERV3 family overexpression has been implicated in numerous cancer types and is associated with tumor size and stage. ERV3 overexpression is seen in other malignancies, including Hodgkin’s lymphoma cells in which down-regulation of ERV3 can be seen in HL cells compared to normal cells [40]. Furthermore, 30% of ovarian cancer patients have antibodies against ERV3, whose expression is undetectable in healthy women [41]. These data suggest ERV protein expression could serve as an important biomarker of patient prognosis. Among the nearby genes that were concordant in their differential expression with their nearby ERVs were RBP7, CADM1, SPINK5 and TUSC3. RBP7 has been identified as a biomarker with strong prognostic ability in colon cancer in addition to contributing to the malignant phenotype of colon cancer cells [42]. The remaining three genes (e.g., CADM1, SPINK5 and TUSC3) were identified as tumor suppressor genes [43,44,45]. As all three genes shared concordant regulation with their nearby ERVs, there is a possible regulatory effect that could affect the suppressive abilities of the natural immune system. We demonstrated that the HPV status had the most profound association with ERV expression, with 3313 ERVs differentially expressed in HPV-positive and- negative subgroups. The nearby affected genes included CLCNKA, ST6GALNAC5, NCAM1, CADM1 and STXBP6, among many others. ST6GALNAC5 is a glycosyltransferase that prevents cell death, and its overexpression in gastric cancer as well as in colon cancer was shown to inhibit apoptosis [46,47]. Interestingly, its knock down resulted in enhanced chemosensitivity in pancreatic cancer cell lines [48]. This gene function or relation to cervical cancer has yet to be discussed in the literature to date. Additionally, NCAM1 encodes a protein that is a member of the immunoglobulin family and has been predominately studied as it portends therapy resistance and cell proliferation in leukemia and lung cancer [49,50]. As it relates to cervical cancer, NCAM1 downregulation has been shown to be associated with carcinoma in situ of the cervix in comparison to its levels in women with HPV infections in the absence of dysplasia [51]. With the unique virally mediated carcinogenic mechanism of most cervical cancers involving the incorporation of HPV E6 and E7 genes into human host DNA, the role of ERVs in this integration of the viral genome and the mechanism by which ERV-mediated epigenetic modulation facilitates carcinogenesis remain unknown; however, our data including ERV expression and nearby genes impacted by these differentially expressed ERVs may allow for further exploration. Furthermore, it is noteworthy that other viruses have been found to be associated with ERV expression in cancer and other diseases, including SARS-Cov-2, DENV-2, KSHV, HBV virus X protein, CMV, HHV-6B, HIV, EBV and HSV-1 [52,53,54,55,56,57,58,59,60].

There are several prognostic factors that have been investigated for their impact on the survival of cervical cancer patients, including histologic subtype (adenocarcinoma, with worse outcomes than squamous cell carcinoma), maximum depth of cervical stromal invasion, longitudinal tumor diameter, lympho-vascular space invasion, parametrial invasion and pelvic lymph node metastases. Multivariate analyses support that, of these, nodal metastases, histology and tumor diameter are independent predictors of survival. These three predictors were analyzed to create prognostic indices, leading to a classification into low-risk, moderate-risk and high-risk groups, more predictive of survival than FIGO stage alone [2]. An alternative database study aimed at a prognostic nomogram creation identified age, race, marital status, tumor grade, FIGO stage, histology, tumor size and nodal status as significant to survival outcomes. These data suggest the superiority of this multi-factorial model to FIGO staging alone [3].

It is well established within head and neck squamous cell carcinomas that the HPV status significantly impacts survival, with HPV-positive tumors being more favorable. A transcriptome analysis by Zhang et al. of squamous cell carcinomas of the cervix suggested that the same applies to cervical cancer, with HPV positivity conferring improved survival, possibly from enhanced DNA repair mechanisms limiting somatic mutation accumulation [61]. The transcriptome data from the analysis by Zhang et al. was applied to our ERV modeling, allowing for a novel reporting of ERV expression in squamous cell carcinomas of the cervix [18]. Our internally validated model which combined the 67-ERV panel expression data and the clinical data, discriminated groups at high risk of recurrence and low-risk of recurrence with far greater superiority than the clinical variables alone. The ability to successfully classify patients as either high-risk or low-risk for disease progression is of incredible value for patient management. In our previous study [11], we demonstrated that a combination of ERV expression signatures alongside clinical data for patients could provide valuable insights for the prediction of a specific patient’s recurrence probability.

In addition to the potential of using ERV expression to improve patient prognostication, our data also introduce novel targets to explore for the treatment of this disease. For example, FDA-approved nucleotide analogs such as 5-Azacytidine (5AzaC) cause multiple dysregulations including inhibition of DNA methyltransferases and can lead to the upregulation of immune signaling through the induction of a viral mimicry status, encouraging the destruction of tumor cells through antiviral immune response pathways, causing an interferon response and apoptosis [4,5,62]. The drug 5AzaC downregulates the HPV E6 and E7 oncogene expression in cervical cancer and head/neck SCC cell lines, and there is evidence that treatment with 5Aza-C can enhance the potency of the HPV DNA vaccine through the upregulation of CRT/E7 expression and the subsequent enhanced E7-specific CD8+ T cell immune response (4). We aim to further investigate whether these drugs restore the activity of the antiviral immune response and DNA repair genes via ERV induction in organoid and xenograft models.

The strengths of this work stem from its novelty in that the exploration of ERV expression in cervical cancer and of the role of ERVs in carcinogenesis and as potential therapeutic targets is yet unexplored. Given the significant improvement in prognostication using clinical factors combined with our 67-ERV panel expression data, we performed an internal validation which lends further strength to the applicability of our findings. We also performed protein expression validation of ERV3. A future study will be performed utilizing an expanded cohort and including the use of The Cancer Genome Atlas, with additional available clinical variables, to further investigate ERV expression differences and affected nearby genes as well as to provide further validation.

In summary, ERV RNA expression in cervical cancer tumors is significantly different depending on racial cohorts, HPV subgroups and disease stages. The correlation of the expression of certain ERVs in cervical cancers alongside clinical factors significantly improves prognostication when compared to using clinical factors alone; ERVs may also serve as future therapeutic targets.

## 4. Materials and Methods

### 4.1. Sample Selection

The NCBI GEO database was accessed with the utilization of GSE151666 [18] for the analysis of cervical cancer with available correlative HPV data, which were collected as part of a tumor banking study at Washington University School of Medicine (IRB: 201105374). As previously described, whole transcriptome sequencing was performed before treatment on primary cervical cancer samples from 68 (FFPE tissues) cervical cancer patients treated at the Washington University School of Medicine [18].

Available clinical information included age at diagnosis, race, tumor histotype, FIGO clinical stage, recurrence and HPV status including HPV-absent or HPV-positive subtypes 16, 18 or a high-risk group including subtypes 31, 33, 58, 59, 45, 52, 56 and 66.

### 4.2. Locating Ancestry from Sequence Reads (LASER)

To identify the geographical ancestry of each patient, we used LASER [63], which estimates an individual’s ancestry by directly analyzing the sequence reads without calling genotypes. The parameters provided to LASER and the methodology can be found in our previous study [11]. Briefly, all patient RNA-seq data were aligned, and the expression of SNPs known to be associated with ancestral differences was quantified. The numeric estimates were processed by LASER to generate ancestral estimations.

### 4.3. RNA-Seq Analysis

Following LASER analysis, the patients of primarily Asian (n = 3) and Hispanic (n = 2) ancestry were removed prior to alignment, resulting in a total of 63 samples that were chosen for downstream analysis. For the purposes of comparing early- and locally advanced-stage samples, one patient (n = 1) was excluded due to not fitting within the appropriate FIGO stage classification. Using a pipeline previously validated in prostate cancer [11], we analyzed and mapped reads from GEO GSE151666 [18] to the human genome reference sequence (hg38) with STAR [64]. Using the *coverage* function from bedtools [65], we provided these alignments and a BED file containing the locations of retroviral elements from the Human Endogenous Retrovirus database (HERVd) [21] as inputs to calculate the number of reads mapped at each ERV locus, thus providing the raw expression count data for the ERVs.

### 4.4. Transcriptome Analysis of ERVs in Cervical Cancer Patients

To determine ERV expression differences between early-stage (FIGO stage 1A1– \1B2) and locally advanced-stage (stage 1B3–IVA) disease, HPV+ and HPV- groups and patients of different ancestries (i.e., Black American = BA and White American = WA) with cervical cancer tumors, the aforementioned raw read count data for the regions annotated by the HERVd were imported for use in DeSeq2 [20]. The analysis was performed as mentioned in our previous work [11]. Briefly, the raw read counts for the ERVs were normalized using the size factor estimates from the host gene counts. Differential expression analysis was conducted using the three conditions (Tumor stage, HPV status, and ancestry).

### 4.5. ERV-Targeted Genes

As ERVs may modulate the activity of adjacent genes [66,67,68], a filtered selection of ERVs nearby genes located within a vicinity of 5000 bp were identified using the bedtools [65] closest function. Filtering was done by selecting ERVs that were found to be differentially expressed in each of the three conditions.

### 4.6. TMA Analysis of ERVs with Protein-Coding Potentials

We obtained 59 cervical cancer TMA cores from NovusBio with the following distribution: normal cervix (n = 4), precancer/dysplasia (n = 4), primary invasive cervical cancer (n = 46) and correlative metastatic nodal tissue (n = 5). The cervical cancer cores were primarily squamous cell carcinomas (45/50), with 2 adenocarcinoma and 3 adenosquamous carcinomas. The samples included 27 Stage I, 1 Stage II, and 19 Stage III tumors according to the FIGO stage distribution.

Blocks from each TMA were stained with antibodies for ERV3 and ERVK-7 by CrownBio (Per CrownBioInc, San Diego, CA, USA). Immunohistochemistry was performed on a Bond RX autostainer (Leica Biosystems, DS9800, Wetzlar, Germany) with heat-induced epitope retrieval (HIER) treatment in EDTA buffer (pH 9.0) using a standard protocol. The primary antibodies used were rabbit polyclonal ERVK-7 antibody (Invitrogen, # PA549515 1:100) and rabbit polyclonal ERV-3 antibody (Invitrogen, PA548577, 1:100). Bond Polymer Refine Detection (Leica Biosystems, DS9800, Wetzlar, Germany) was used as a secondary antibody detection system according to the manufacturer’s standard protocol. After staining, the sections were dehydrated and film-coverslipped using a TissueTek-Prisma and Coverslipper (Sakura Finetek, Torrance, CA, USA). Whole-slide scanning (40×) was performed on a NanoZoom Digital Slide System NDP2.0-HT (Hamamatsu, Shizuoka, Japan).

### 4.7. Statistical Analysis of ERV Protein Expression in TMAs from Early- and Locally Advanced-Stage Cervical Cancer Patients

From the original set of 50 TMA cores containing cancerous tissues, 3 cores that lacked FIGO stage data were excluded from the analysis. Samples with a FIGO Stage of IA1–1B2 cancer were classified as early-stage, while those of FIGO Stage 1B3-IVA disease were classified as locally advanced-stage [69]. As a result, 19 cores containing early-stage and 28 cores containing locally advanced-stage cancer tissue were quantified using QuPath [23]. Positive Pixel counts were computed for each individual core to quantify the amount of stained tissue on the slides as described previously [11]. Briefly, the percent positivity was calculated by dividing the number of positive pixels detected by the total number of pixels identified for each stain, excluding the non-stained background regions. A Student’s *t*-test was used for comparison between the early- and the locally advanced-stage TMA data. A *p*-value < 0.05 was considered significant.

### 4.8. Survival Analysis

All-cause survival time was calculated from the date of diagnosis to the date of death or, if censored, the date of the last contact or visit. Survival time was considered to be right-censored if the patient was lost to follow-up or alive at the end of the study period. Panels of discriminatory ERVs and risk determination were identified by using the same methodology documented in our previous study [11]. Briefly, a pre-filtering step using univariate Cox proportional hazards models was conducted on all ERVs. The most significant ERVs selected from this pre-filtering step were provided as an input to a new Lasso-Cox proportional hazards model, which selected the ERVs of interest [24]. These expression data of these new ERVs were provided alongside the clinical data to create risk prediction scores for each sample. The risk evaluations were used to generate a Kaplan–Meier survival model. Internal cross validation using a leave-one-out approach was conducted to validate the models.

## Figures and Tables

**Figure 1 ijms-24-01492-f001:**
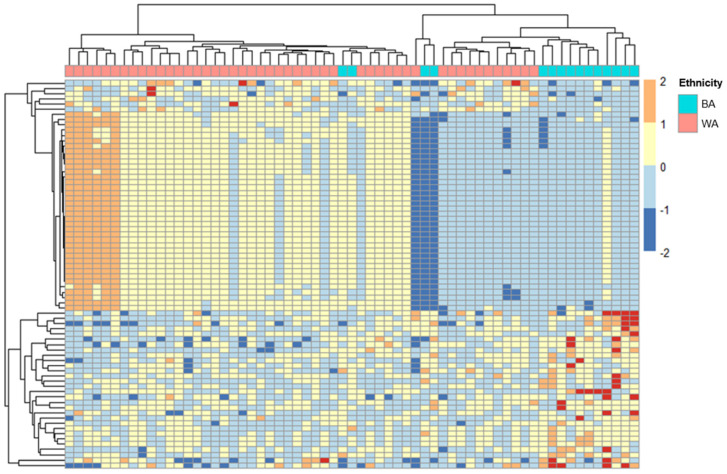
Transcriptome analysis of ERVs in Black American and White American cervical cancer patients. ERVs from 63 cervical cancer patients were mapped to the HERVd database. Differential expression analysis and hierarchical clustering analysis (Euclidean) were performed in DeSeq2 to organize the tumor samples based on their ERV transcription profiles and the ancestry of the cervical cancer patients. The heat map was generated using the normalized expression values for 74 significantly differentially expressed ERVs in BA vs. WA. The plot above depicts the most significant differentially expressed ERVs (*p_adj_* < 0.1) in BA (n = 15) vs. WA (n = 48) that were differentially expressed among samples of different ancestries (a complete list of differentially expressed ERVs can be found in Appendix A). Each column represents one sample, and each row represents a single ERV. ERV expression level is indicated by color intensity (−2.0 to 2.0) on a log scale; red indicates ERVs with higher expression, and blue indicates those with lower expression. ERV expression was normalized to the mean of all samples. In the dendrogram on the top, the ancestral origin of each patient is indicated by color. Red indicates patients of predominantly European/Middle Eastern (n = 48) geographical ancestry, while blue represents patients of African ancestry (n = 15). BA = Black American; WA = White American.

**Figure 2 ijms-24-01492-f002:**
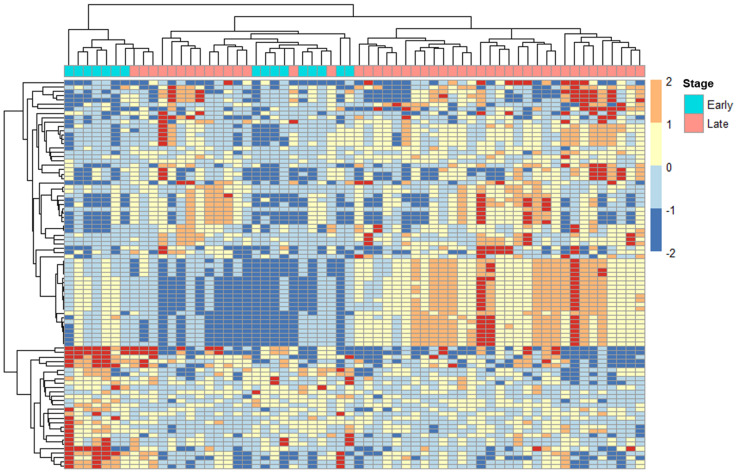
Transcriptome analysis of ERVs in early and locally advanced-stage/late cervical cancer patients. Heatmap generated using the normalized expression values for ERVs from the Human Endogenous Retrovirus database (HERVd). The plot above depicts the most significant ERVs (*p_adj_* < 0.1) that were differentially expressed among tumor samples of different stages; late/locally advanced patients (n = 46), early patients (n = 16), 1 patient was excluded from this comparison (a complete list of differentially expressed ERVs is displayed in Appendix A). ERV expression levels is indicated by color intensity (−2.0 to 2.0) on a log scale; red indicates ERVs with higher expression, and blue indicates ERVs with lower expression. Early = FIGO early stage IA1–1B2; Late = FIGO locally advanced stage IB3-IVA

**Figure 3 ijms-24-01492-f003:**
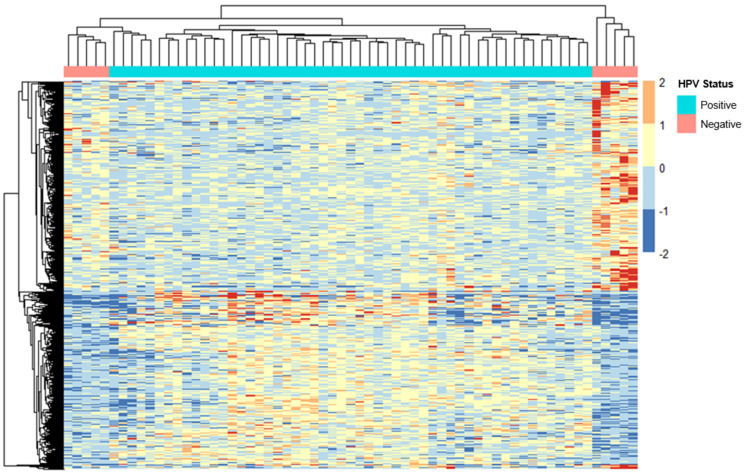
Transcriptome analysis of ERVs in HPV-positive and -negative cervical cancer patients. Heatmap generated using the normalized expression values for ERVs from the Human Endogenous Retrovirus database (HERVd). The plot above depicts the most significant ERVs (*p_adj_* < 0.1) that were differentially expressed among samples with differing HPV diagnoses. The plot above depicts the most significant ERVs (*p_adj_* < 0.1) that were differentially expressed among samples of different HPV status, i.e., positive (n = 53) and negative (n = 10). A complete list of differentially expressed ERVs is displayed in Appendix A. ERV expression is indicated by color intensity from −2.0 to 2.0 on a log scale; red indicates ERVs with higher expression, and blue indicates ERVs with lower expression. HPV = human papillomavirus.

**Figure 4 ijms-24-01492-f004:**
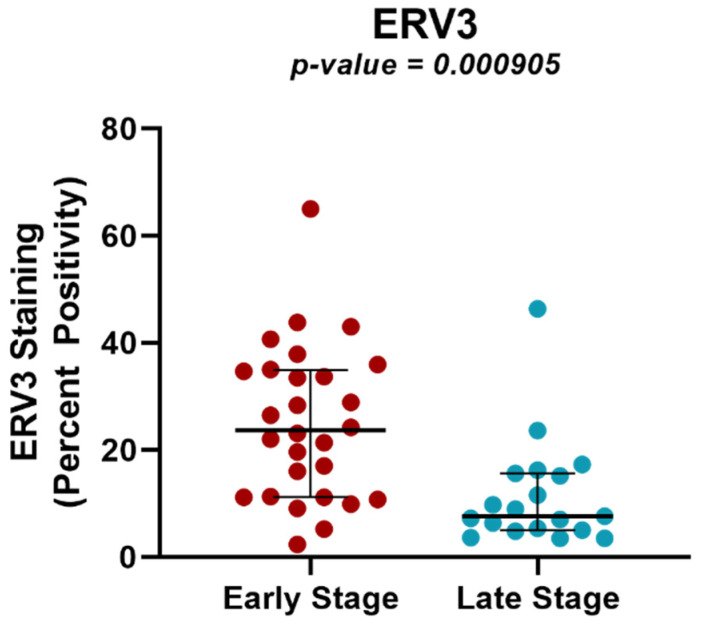
Differential ERV3 expression by immunohistochemistry (IHC) between early- and locally advanced-stage cervical cancers. Scatter plot representing the IHC results of a tissue microarray of cervical cancer samples stained with ERV3 antibodies. ERV3 expression is significantly different between the early-stage (n = 28) and late-stage (n = 19) cohorts (*p* = 0.000905). The y-axis represents the percentage of tumor cells that are positive for ERV3. Early = FIGO early stage IA1–1B2; Late = FIGO locally advanced stage IB3-IVA.

**Figure 5 ijms-24-01492-f005:**
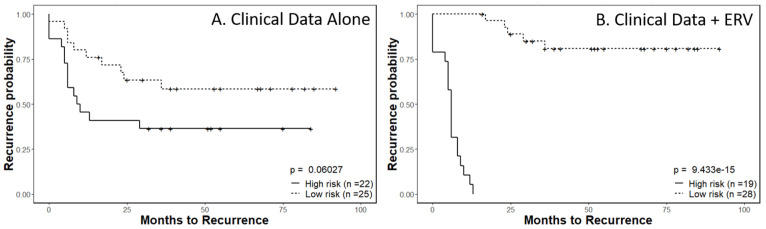
Kaplan–Meier survival plots of recurrence risk for cervical cancer patients. Kaplan–Meier survival analysis of recurrence using low-risk or high-risk models based on (**A**) clinical factors only (*p* = 0.06027) compared to (**B**) a combination of clinical factors and 67-ERV RNA panel expression levels (*p* = 9.433 × 10^−15^). The analysis was performed on a test set of n = 47 patients selected from the total cohort of 63 patients.

**Table 1 ijms-24-01492-t001:** Summary statistics for all samples filtered from GSE151666. Patients of Asian or Hispanic ancestry were excluded from the downstream analysis.

Characteristic	N = 63 (%)
**Age**	53 years (46–64 years)
**Race**	
White American	48 (76%)
Black American	15 (24%)
**FIGO Stage**	
Early: IA1–IB2	16 (25%)
Late/locally advanced: IB3–IVA	46 (75%)
Excluded: IVB	1 (<1%)
**HPV Status**	
HPV positive	53 (84%)
HPV negative	10 (16%)

HPV = human papilloma virus.

## Data Availability

Data used for RNA-seq analysis can be found at GEO Accession GSE151666. The other datasets used/generated in this study are available from the corresponding authors on reasonable request.

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
