# Peer review of "Endogenous Retrovirus RNA Expression Differences between Race, Stage and HPV Status Offer Improved Prognostication among Women with Cervical Cancer"

_ijms, 2023, doi:10.3390/ijms24021492_

Round 1

Reviewer 1 Report

Human retroelements (HERVs) are retroviral origin sequences fixed in the human genome. Recent studies indicate that HERVs are associated with many diseases, including cancers. In this work, the authors found that ERV RNA expression differences in cervical cancer tumors was significantly different 31 among racial cohorts, HPV-subgroups, and disease stages. This is very interesting. However minor work is required in the discussion.

1. HPV status had the most profound associated with ERV expression. In fact, many other viruses, including SARS-Cov-2(EBioMedicine 2021 Apr; 66:103341), DENV-2 (Virology 2020a;544:21), KSHV(Oncogene 2018; 37:4534), HBV(Virus Genes 2017 Dec; 53(6):797-806, CMV( J Clin Virol 2015;68:28), HHV-6B(J Clin Virol 2009; 46:15), HIV(AIDS Res Hum Retroviruses 2007;23:116), influenza A (Retrovirology 2006; 3:44), EBV(J Virol 2004; 78:7852, HSV-1( Mol Cells 2003; 15:75), have been found to be associated with ERVs expression. The authors should discuss this in the discussion.

2. In the work, ERV expression signatures in 37 tumor biopsies may therefore be useful to help identify patients at greater risk of recurrence. In fact, there are many studies show that HERVs, as biomarkers, participate in the development of different cancers, including Prostate cancer(Viruses. 2021;13(3):449.), hepatocellular carcinoma(Cell Death Discov. 2021;7(1):177.), renal cell carcinoma(JCI Insight. 2018; 3(16):e121522.), ovarian cancer(Clin Cancer Res. 2015;21(2):471483), urothelial cell carcinoma (Oncogene 2014;33(30):39473958.), breast cancer(Int J Cancer. 2014;134(3):587595.), testicular cancer(Nucleic Acids Res. 2010;38(7):22292246), and endometrial carcinoma(J Mol Med.2006;85(1):2338). The authors should discuss this in the discussion. 

Reviewer 2 Report

review enclosed
